# United Yet Distinct: Domain Preservation via Divergence Reduction

## Abstract

Although there is a vast amount of data available for training Large Language Models (LLMs), data privacy concerns can limit centralized data aggregation, therefore limiting the learning capacity of LLMs on data from distributed sources. Federated Learning (FL) has emerged as a dominant framework for distributed training. The objective of FL is to preserve privacy while improving the performance of participating clients. However, the non-IID nature of participating clients can degrade model performance. Parameter Efficient Fine-Tuning (PEFT) enables adapting LLMs to downstream tasks with minimal parameter additions and updates to their existing parameters. Preserving performance while learning from data in a distributed setting warrants the need for efficient training frameworks that can enable LLMs to learn from disparate data. In this paper, we design and propose a novel FL aggregation algorithm, Divergence Reduction in Federated Training (DRIFT), which accounts for the divergence between clients during model aggregation and disseminates custom aggregated parameters back to each client. DRIFT measures the degree to which the PEFT parameters of the participating clients diverge and takes advantage of the graph-based structure implied by this divergence. We design two variants of DRIFT and, through extensive experimentation, show how DRIFT outperforms well-established baselines. Our training data and code are available at: https://anonymous.4open.science/r/drift-240F.

## 1 Introduction

The diversity of tasks performed by Large Language Models (LLMs) makes them an appealing tool for building intelligent applications capable of performing mundane to more specialized tasks. In particular, LLMs have been shown to simulate reasoning as a chained sequence of outputs leading to a desired outcome (Raj et al., 2025; Liu et al., 2024a; Grattafiori et al., 2024; Yang et al., 2024; Team et al., 2023; Achiam et al., 2023; Wei et al., 2022; Brown et al., 2020). LLMs can range in size from a few million to billions of parameters. Consequently, training and fine-tuning LLMs can be computationally prohibitive and expensive. Low-Rank Adaptation (LoRA) (Wang et al., 2024a; Kwon et al., 2024; Chen et al., 2024b; Guo et al., 2024; Hu et al., 2022) has emerged as a compelling paradigm to efficiently fine-tune LLMs. LoRA injects trainable low-rank weight matrices into the existing layers of the LLM architecture. This significantly reduces the cost of training while improving performance on downstream tasks. Federated Learning (FL) has proven to be a useful privacy-preserving framework for distributed model training (Wu et al., 2025; Ye et al., 2024; Zheng et al., 2024; Che et al., 2023). Without sharing data, participating clients only perform local model updates and share updated parameters with a centralized server. The server performs aggregation and distributes a global model to the clients (Reddi et al., 2021; Li et al., 2020; McMahan et al., 2017). FL is particularly useful when data sources are distributed and disparate, and privacy preservation is paramount. However, for LLMs, due to the inherent difference in logic, the task of reasoning varies by domain (Lee et al., 2025; Sun et al., 2023). Therefore, aggregating models under centralized FL can cause performance degradation due to divergence in reasoning chains that can result from the domain specificity of different clients (Kyllonen, 2020; Elsabbagh & Karmiloff-Smith, 2006; Liang et al., 2024).

Motivated by this phenomenon, we propose a client aggregation mechanism that allows participating clients to benefit from each other while maintaining their local characteristics. Specifically, our framework trains

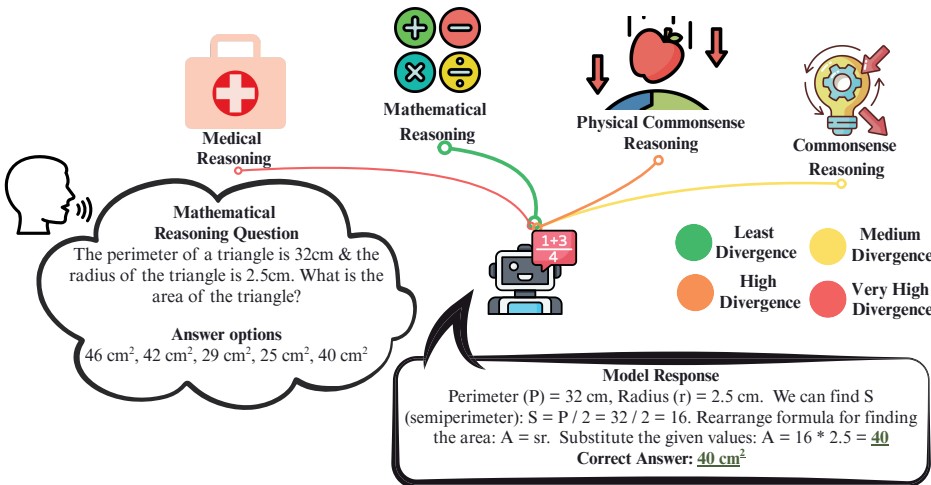

Figure 1: A client benefits from other less divergent participating clients. The connections in the figure show the degree of similarity between a client and other clients.

LLMs in an FL setting so that they benefit from less divergent participating clients, mitigating the performance decline that can result from client heterogeneity. As shown in Figure 1, a given client benefits the most from another client with similar characteristics. For centralized FL aggregation, we design a novel server-side aggregation algorithm, Divergence Reduction In Federated Training (DRIFT). DRIFT measures the degree of divergence and builds graph-like structures between participating clients. Then it uses graph search algorithms to perform custom server-side aggregation for each participating client. At the end of an FL round, each client receives aggregated parameters specific to its characteristics. We design two variants of DRIFT, based on Shortest Path (SP) and Minimum Spanning Tree (MST) graph search problems, DRIFT SP and DRIFT MST, respectively, and use Chain of Preference Optimization (CPO) (Zhang et al., 2024) in conjunction with LoRA for local training. Furthermore, we conduct model training and evaluation across a diverse range of reasoning tasks/domains covered through 8 datasets. We posit that, through custom aggregation, each client benefits as long as the parameters of the aggregated clients do not drastically diverge from each other. The main contributions of our paper are:

- Design and implementation of a novel server-side centralized FL aggregation algorithm, for LLM training.

- Integrating our algorithm with cutting-edge training methods for LLMs and extending FL aggregation to graph-search algorithms.

- Extensive experiments on a diverse set of 8 different natural language datasets with established FL baselines using Llama 3.1 8B and Qwen 2.5 7B as base models.

## 2 Preliminary

### 2.1 Federated Learning (FL)

In FL multiple clients participate in distributed training and share model parameters or gradients with the server. The server implements an aggregation algorithm and distributes the aggregated parameters to the clients. Formally given $K$ clients and the total dataset as $D_K = \{D_1, D_2, .., D_k\}$ where $D_k = \{x^{(i)}, y^{(i)}\}_{i=0}^{N}$ denotes the local dataset for a client, weighting each client by its local sample size, the FL objective (McMahan et al., 2017; Li et al., 2020) is:

$$\min_w f^*(w) \triangleq \mathbb{E}_{p_k}[f_k(w)] \tag{1}$$

Here, $w$ represents local model parameters, $p_k = \frac{|D_k|}{|D_K|}$ the proportion of client samples, and $f_k(w) \overset{\Delta}{=} \frac{1}{|D_k|}\mathcal{L}_k(w, D_k)$ the empirical loss of the client. Table 1 provides the notational symbols used in the paper.

## 2.2 Preference Optimization

Direct Preference Optimization (DPO), through the construction of preference pairs, further enables the fine-tuning of an LLM by aligning it with preferred responses (Rafailov et al., 2023). Given a language model $\pi_\theta$, parameterized with $\theta$, prompt $x$ and a labeled human preference dataset $D = \{x^{(i)}, y_w^{(i)}, y_l^{(i)}\}_{i=0}^N$, the DPO objective, given in Equation 2, is to implicitly learn a reward function such that it maximizes the probability of preferred generations.

$$\mathcal{L}_{\text{DPO}}(\pi_\theta, \pi_{ref}) = -\mathbb{E}_{(x,y_w,y_l)\sim\mathcal{D}}[\log\sigma(\beta\frac{\pi_\theta(y_w|x)}{\pi_{ref}(y_w|x)} - \beta\frac{\pi_\theta(y_l|x)}{\pi_{ref}(y_l|x)})] \tag{2}$$

Chain of Preference Optimization (CPO) (Zhang et al., 2024) uses an LLM ($\pi_\theta$) and Tree of Thoughts (ToT) Yao et al. (2023) to generate a final response by reasoning with intermediate thoughts $[z_1, z_2, .., z_i]$. Subsequently, it builds preference pairs from the intermediate thoughts for DPO. Each intermediate thought, $z_i$, is generated such that $z_i = \pi_\theta(x|s_{i-1})$, where $x$ is the prompt and $s_{i-1} = z_1, z_2, .., z_{i-1}$ represents the previously generated thoughts. Through pruning, preference pairs are created such that $\pi_\theta(z_i^w|x, s_{i-1}^w)$ is the probability of generating preferred thoughts and $\pi_\theta(z_i^l|x, s_{i-1}^l)$ is the probability of generating dispreferred thoughts. The CPO objective is defined as:

$$\mathcal{L}_{\text{CPO}}(\pi_\theta, \pi_{ref}) = -\mathbb{E}_{(x,z^w,z^l,s^w,s^l)\sim\mathcal{D}}[\log\sigma(\beta\frac{\pi_\theta(z_i^w|x, s_{i-1}^w)}{\pi_{ref}(z_i^w|x, s_{i-1}^w)} - \beta\frac{\pi_\theta(z_i^l|x, s_{i-1}^l)}{\pi_{ref}(z_i^l|x, s_{i-1}^l)})] \tag{3}$$

## 3 Problem Setup

We apply our proposed method to LLMs, particularly in the context of Parameter Efficient Fine-Tuning (PEFT) (Han et al., 2024). Low-rank Adaptation (LoRA) (Hu et al., 2022) is a PEFT method that enables training LLMs on downstream tasks with minimal additions to existing parameters. Given a fixed weight matrix $W_0 \in \mathbb{R}^{m\text{x}n}$, LoRA constrains the update $W_0 = W_0 + \Delta W$ by introducing two reduced rank matrices $B \in \mathbb{R}^{m\text{x}r}$ and $A \in \mathbb{R}^{r\text{x}n}$. Here $\Delta W = BA$ and $r << min(m, n)$ is the rank of LoRA. For a given LLM parameterized with $\Phi$, LoRA learns a set of parameters $\Theta$ such that $|\Theta| << |\Phi|$. Our method exploits this low-dimensional property to distinctly aggregate parameters for each client by minimizing the divergence to all

Table 1: Summary of main notational symbols.

| Notation | Definition |
|---|---|
| $K$ | Number of total clients. |
| $D_k : \{x^{(i)}, y^{(i)}\}_{i=0}^N$ | Local dataset for the $k^{th}$ client. |
| $x^{(i)}$ | Prompt for the LLM. |
| $z_i^w; z_i^l; s_{i-1}^w$ | Preferred thoughts under CPO; Dispreferred thoughts under CPO; Set of preferred thoughts leading up to $z_i^w$. |
| $\Theta; \Theta_k$ | Set of LoRA parameters for a given set of clients; LoRA parameters for the $k^{th}$ client. |
| $w$ | Model parameter weights. |
| $\mathcal{L}_k$ | Empirical loss of the $k^{th}$ client |
| $d_{s,t}$ | Divergence between a pair of clients, connecting a source (s) client to a target (t) client. |
| $P; P_t; p$ | A set of shortest paths for a given client; $t^{th}$ shortest path; a single path containing a set of divergences between adjacent clients. |
| $\mathcal{G}$ | Graph generator function. |
| $\|\rho\|_1$ | L1 Norm of the weight vector $\rho$. |
| $\hat{\rho}$ | Normalized weight vector containing weights assigned to each client. |
| $G(V, E)$ | Client graph with $V$ vertices and $E$ edges, representing all clients. |
| $\delta$ | Divergence threshold. |
| $B; b$ | A set of batches for local training; 1 training batch. |
| $\eta$ | Local learning rate. |
| $E; e$ | Number of local epochs; 1 local epoch. |
| $d_{s,t}^{-1}$ | Multiplicative inverse ($\frac{1}{d_{s,t}}$) of $d_{s,t}$. |

**Traditional Federated Learning**        **DRIFT**

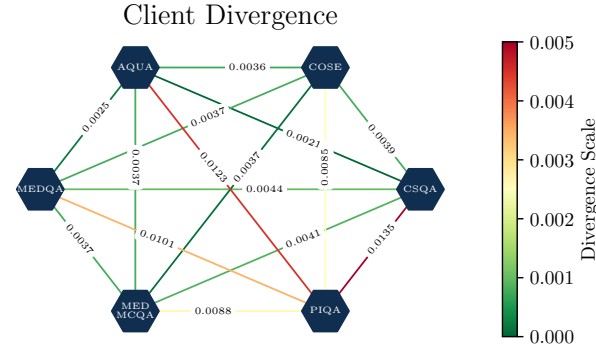

Figure 2: Traditional vs. DRIFT FL aggregation. In DRIFT, the server measures divergence between clients and distinctly aggregates models for each client.

other clients. Formally, given a client $k \in K$, let the subset of least divergent client parameters for the current FL round be denoted as $\Theta = \{\Theta_1, \Theta_2, \Theta_3, \cdots, \Theta_s\}$, where $s \subseteq K$. Furthermore, let $\rho = \{\rho_1, \rho_2, \rho_3, \cdots, \rho_s\}$ be the set of weights assigned to each client in $\Theta$. Then the local objective for client $k$ is given in Equation 4 and the parameter update, in a given FL round, is given in Equation 5.

$$\min_{\Theta_k \in \mathbb{R}} f(\Theta_k) = \frac{1}{|D_k|} \mathcal{L}_k(\Theta_k, D_k) \tag{4}$$

$$\Theta_{k_{t+1}} = \rho_k \Theta_k + \sum_{i \in s} \rho_i \Theta_i \tag{5}$$

Here, $D_k = \{x, z_i^w, z_i^l, s_{i-1}^w\}$ is the chain of preference thoughts dataset created using CPO and $\mathcal{L}_k$ is the empirical local loss of the client.

## 4 Proposed Method

As shown in the workflow diagram in Figure 2, in our framework, each client receives distinctly aggregated LoRA parameters based on the divergence from other participating clients. In each FL round, the server measures the degree of divergence between each pair of participating clients, creating a graph-like structure between them, as shown in Figure 3. The graph $G(V, E)$ representing $k$ clients is created such that $V$ is a set of vertices representing each client, where $|V| = k$ is the total number of clients in the current FL round. The weight function $d : E \mapsto \mathbb{R}^+$ maps the edges to real-valued weights that determine the degree of divergence between clients. Given an edge $e \in E$, a source client, and a target client vertex, $s, t \in V$, we define the divergence between two distinct clients as:

$$d_{s,t} = SKL(\Theta_s || \Theta_t) \tag{6}$$

Figure 3: Client divergence graph. Each node represents a client with its own distinct dataset, and edge weights represent the divergence between clients.

Here, *SKL* is the Symmetric KL divergence (Huang et al., 2015; Pu et al., 2017; Chen et al., 2018; Andriamanalimanana et al., 2019; Ruiz & Titsias, 2019; Yao & Liu, 2025) defined as:

$$SKL(\Theta_s || \Theta_t) = KL(\Theta_s || \Theta_t) + KL(\Theta_t || \Theta_s) \tag{7}$$

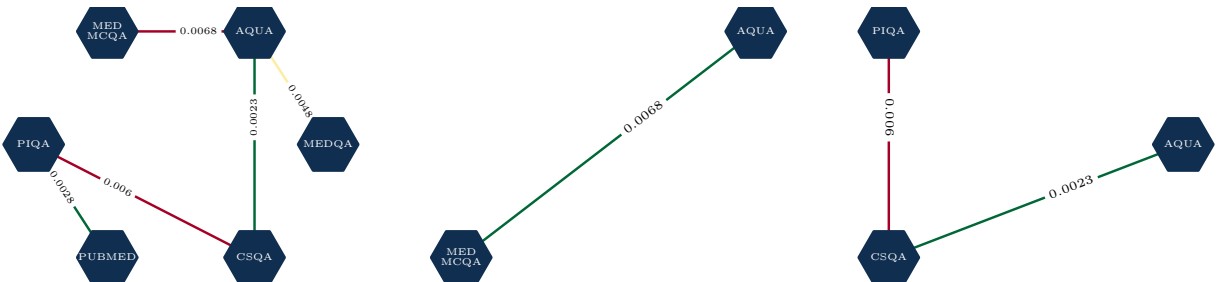

Figure 4: DRIFT MST minimum spanning tree of participating clients and immediate neighbors.

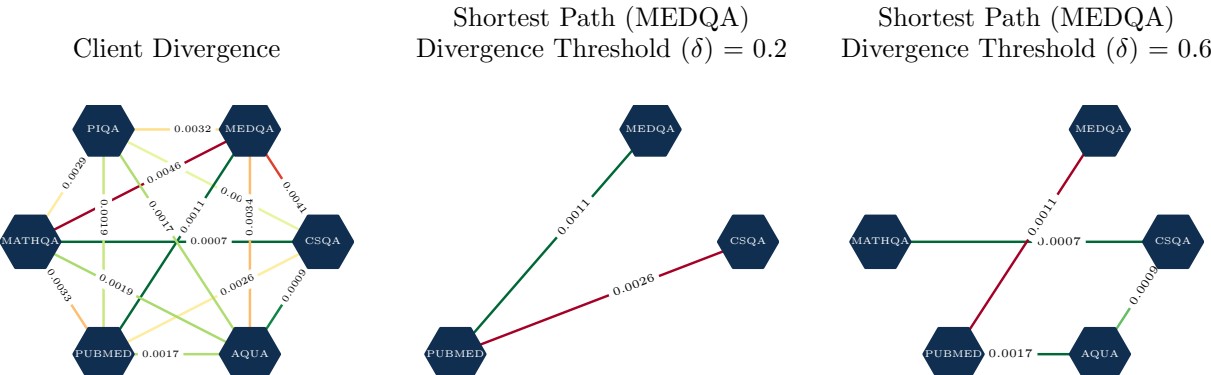

Figure 5: DRIFT SP. The client graph shows divergences between all clients. Shortest path plots show client (MEDQA) aggregation with different divergence thresholds ($\delta$). Higher $\delta$ corresponds to aggregation with more clients.

The KL divergence (Goodfellow et al., 2016), from a source client to a target client, with model parameter weights $w$ and probability distribution $p$, is further defined as:

$$KL(\Theta_s||\Theta_t) = \sum_{w \sim p_{\Theta_s}} \log p_{\Theta_s}(w) \frac{p_{\Theta_s}(w)}{p_{\Theta_t}(w)} \tag{8}$$

KL divergence has a property of non-negativity, but it is non-symmetric from a source distribution to a target distribution. However, Symmetric KL divergence is symmetric from a source distribution to a target distribution (Kullback, 1997). Given these properties, we can ensure that the edge weights between the client vertices are non-negative and symmetrical. Using Equations 6, 7, and 8, we compute the divergence between each source and target client pair in the current FL round as:

$$\{d_{1,2}, d_{1,3}, \cdots, d_{k-s,k-t+1}, d_{k,k}\} \ s.t. \ d_{s,t} \mapsto \mathbb{R}^+, \forall s,t \in k \tag{9}$$

Following the convention in Cormen et al. (2022), we formulate client aggregation, under DRIFT, as a Minimum Spanning Tree (MST) and a Shortest Path (SP) problem and propose two aggregation schemes, DRIFT MST and DRIFT SP, respectively. Shortest Path (SP) and Minimum Spanning Tree (MST) lend particularly well to our framework as the objective of each graph search problem enables finding a path of least divergence between a set of clients.

### 4.1 DRIFT MST

Given a client graph $G(V, E)$, we find a subset of edges $T \subseteq E$, connecting all clients, that minimizes the total divergence between clients given by:

$$\min d(T) = \sum_{(s,t) \in T} d(s,t) \tag{10}$$

The client-specific aggregation under DRIFT MST is done by aggregating a given source client with the target clients that are its immediate neighbors (Chen et al., 2002). This is shown in Figure 4.

### 4.2 DRIFT SP

Given a client graph $G(V, E)$ and a path, $s \overset{p}{\rightsquigarrow} t$, let $p = \{d_1, d_2, d_3, \cdots, d_k\}$ be a set of weights of constituent edges from a source client to a target client. The shortest path between a source client and a target client is a path with minimum total divergence given by:

$$\min d(p) = \sum_{i=1}^{k} d(v_{i-1}, v_i) \tag{11}$$

Since, $G(V, E)$ is a fully connected graph of participating clients, Equation 11 gives a set of shortest paths between each source and target client. To select a desired path and to facilitate the exploration vs. exploitation dilemma, we implement a divergence threshold term, $\delta$, that selects the desired path based on the fraction of clients to be included in a given shortest path. Formally, given a set of shortest paths, $P = \{p_1, p_2, \cdots, p_t\}$, from a source client, the normalized path lengths are given as:

$$\hat{P} = \{\frac{|P_i|}{\sum_{i=1}^{t} |P_i|} : P_i \in P\} \tag{12}$$

The selected shortest path is defined as:

$$P_s := \{P_s \in P : \hat{P}_s \leq \delta \;\; and \; \delta \in [0,1] \mapsto \mathbb{R}\} \tag{13}$$

As shown in Figure 5, for selected paths of equal length, we select the shortest path with the least total divergence.

### 4.3 Weighting Clients

Using Equations 10, 11, we can apply well-known minimum spanning tree and shortest path algorithms to get a set of edge weights, $p = \{d_1, d_2, \cdots, d_k\}$, representing the divergence between adjacent clients. Since the magnitude of divergence is analogous to the dissimilarity between a source and a target client; during aggregation, each client is weighted with the normalized multiplicative inverse of its corresponding divergence. This ensures that clients that are less divergent from a source client are assigned higher weights. Given that the source client has the least divergence from itself, we assign it the minimum divergence from the given set of edge weights. Let $d_0^{-1} = \frac{1}{min(p)}$ and $\rho = \begin{bmatrix} d_0^{-1} & d_1^{-1} \cdots d_k^{-1} \end{bmatrix}$ be the weight assigned to the source client and the final set of weights assigned to each client, respectively. The weights used for aggregating clients are then computed as:

$$\hat{\rho} = \frac{\rho}{||\rho||_1} \tag{14}$$

## 5 Analysis

Given that a source client is aggregated with the least divergent clients in a given path, our analysis aims to answer questions regarding the performance bound induced by the number of clients in this path, as well as the nature of the clients contained in the shortest path. Our analysis is based on data heterogeneity assumptions with respect to Non-IID clients, common in FL (Mishchenko et al., 2025; Hamidi & YANG, 2024; Vardhan et al., 2024; Li et al., 2020). Specifically, our objective is to answer the following questions:

I. *Is there a performance bound based on the number of clients that exist in the selected path used for aggregation?*

II. *Does the shortest path to a given client determine less divergent clients to aggregate with?*

## 5.1 Performance Bound

**Assumption 1.** *In a Non-IID setting, client aggregation can degrade model performance.*

**Proposition 2.** *Let $P = \{p_1, p_2, \cdots, p_t\}$ be the set of shortest paths from a source client to all other clients, and let $\hat{P}$ be the normalized path lengths. Using Equation 12 we have:*

$$\hat{P} \sum |P_i| = \{|P_i| : P_i \in P\}$$

*The path with the maximum number of clients is given as:*

$$\max(\hat{P}) \sum |P_i| = \max\{|P_i| : P_i \in P\}$$

*Similarly, the path with the minimum number of clients is given as:*

$$\min(\hat{P}) \sum |P_i| = \min\{|P_i| : P_i \in P\}$$

**Theorem 3.** *Given a divergence threshold $\delta \in [0, 1] \mapsto \mathbb{R}$, the performance bound for a client is determined by the length of the selected path used for aggregation, where $\Omega = \max\{|P_i| : P_i \in P\}$ denotes the maximal path and $\omega = \min\{|P_i| : P_i \in P\}$ denotes the minimal path.*

$$\omega \leq \delta \sum |P_i| \leq \Omega$$

*Proof.* Let $\Omega = \max\{|P_i| : P_i \in P\}$, $\omega = \min\{|P_i| : P_i \in P\}$, and $\hat{P}_s$ be the normalized path length of the selected path, then:

$$\max\{|P_i| : P_i \in P\} \geq \delta \sum |P_i| \quad \forall \delta \leq \hat{P}_s$$

$$\min\{|P_i| : P_i \in P\} \leq \delta \sum |P_i| \quad \forall \delta \geq \hat{P}_s$$

$$\min\{|P_i| : P_i \in P\} \leq \delta \sum |P_i| \leq \max\{|P_i| : P_i \in P\}$$

$$\omega \leq \delta \sum |P_i| \leq \Omega$$

$\square$

This shows that as $\delta$ increases, the length of the path selected for aggregation increases, presenting an exploration vs. exploitation dilemma. In a Non-IID setting, a lower $\delta$ corresponds to a client aggregating with fewer distinct clients. However, a higher $\delta$ would lead to aggregation with more clients having distinct characteristics and potentially degrade model performance. We conducted a parameter study for $\delta$ and validated our analytical findings through experimental results in Table 5.

## 5.2 Client Aggregation

**Assumption 4.** *Aggregating a source client with fewer Non-IID/divergent clients improves model performance.*

**Theorem 5.** *The shortest path aggregates a source client with fewer divergent clients.*

*Proof.* Let $d(i, k)$ be the divergence of client $k$ from client $i$. Then for any edge $(v, k)$ connecting clients $v$ and $k$, by the triangle inequality we have the following:

$$d_{i,k} \leq d_{i,v} + d_{v,k}$$

Using the shortest path, a source client aggregates only with those clients that form the least divergent set. $\square$

---

**Algorithm 1: DRIFT**

---

**Inputs:** $E$ (local train epochs), $B$ (local batch size), $T$ (FL rounds), $K$ (total clients), $C$ (ratio of clients participating in each round), $D_K : \{D_1, \cdots, D_k\}$ (client datasets), $\mathcal{G}$ (graph generator), $MST$ (minimum spanning tree algorithm), $SP$ (shortest path algorithm), $\delta$ (divergence threshold), *variant* ($MST$ or $SP$).

**Server:**

1: initialize $\Theta_0$                                                      ▷ Initialize parameters

2: **for** round $t = 1, 2, \cdots, T$ **do**

3:     $\Theta \leftarrow \emptyset, \hat{\Theta} \leftarrow \emptyset$                                         ▷ Initialize parameter sets

4:     $S_t \leftarrow$ sample $C * K$ clients

5:     **for** k $\in S_t$ **in parallel do**

6:         $\Theta_{k,t+1} \leftarrow$ *ClientUpdate(k, $\Theta_t$)*

7:         $\Theta \leftarrow \Theta \cup \{\Theta_{k,t+1}\}$

8:     **if** *variant* $= MST$ **then**

9:         $G(V, E)^1 \leftarrow MST(\mathcal{G}(\Theta))$

10:     **if** *variant* $= SP$ **then**

11:         $G(V, E)^2 \leftarrow SP(\mathcal{G}(\Theta))$

12:         $G(V, E) \leftarrow$ Compute using 12, 13, and $\delta$

13:     **for** $k \in V$ **do**                                                ▷ For each client

14:         $\hat{\rho} \leftarrow$ Compute using Equation 14

15:         $\Theta_{k_{t+1}} = \rho_k \Theta_k + \sum_{i \in s} \rho_i \Theta_i$                     ▷ $S \subseteq K$ neighbors of client $k$

16:         $\hat{\Theta} \leftarrow \hat{\Theta} \cup \{\Theta_{k_{t+1}}\}$

**ClientUpdate**(k, $\Theta_k$): ▷ for $k^{th}$ client

1: $B \leftarrow \{$Create batches of size $B \in D_k\}$

2: **for** $e = 1, 2, 3 \cdots$ **in** $E$ **do**

3:     **for** $b$ **in** $B$ **do**

4:         $\Theta_k \leftarrow \Theta_k - \eta \nabla \mathcal{L}_k(\Theta_k; b)$

5: **return** $\Theta_k$ to server

---

[a] Create client graph using Equations 6, 7, 8, 10

[b] Create client graph using Equations 6, 7, 8, 11

This shows that, in a Non-IID setting, the shortest path enables client aggregation with least divergent clients, allowing it to preserve its parameter distribution.

# 6 Algorithm

The DRIFT algorithm is presented in Algorithm 1. Our algorithm follows a standard FL setup in terms of communication between the server and the clients. Using LoRA parameters initialized at the server, each client performs an update on its local dataset $D_k$ and communicates the updated parameters to the server. During aggregation, the server creates a client graph $G(V, E)$ based on the divergence between each client. Furthermore, it implements graph search algorithms to identify clients to aggregate with a source client and performs aggregation. The aggregated client-specific LoRA parameters are then communicated to each participating client.

# 7 Experiments

Our experimental setup consisted of 8 datasets, covering commonsense reasoning - CSQA (Talmor et al., 2019), COSE (Rajani et al., 2019), physical commonsense reasoning - PIQA (Bisk et al., 2020), medical reasoning - PUBMEDQA (Jin et al., 2019), MEDQA (Jin et al., 2021), MEDMCQA (Pal et al., 2022), and mathematical reasoning - AQUA (Ling et al., 2017), MATHQA (Amini et al., 2019). For each question, we generated reasoning trees based on Tree of Thoughts (ToT) (Yao et al., 2023), with depth 3 and 2

Table 2: Best rewards by FL method.

| Model | Method | Reward | | | | | | | |
|-------|--------|--------|--------|--------|--------|--------|--------|--------|--------|
| | | AQUA | COSE | CSQA | MATHQA | MEDMCQA | MEDQA | PIQA | PUBMEDQA |
| | FedAvg + LoRA | 4.14 ± 1.12 | 11.23 ± 0.83 | 10.70 ± 1.12 | 3.37 ± 2.66 | 5.55 ± 0.24 | 1.14 ± 0.12 | 9.66 ± 0.22 | 12.23 ± 0.87 |
| | FedProx + LoRA | 4.08 ± 1.22 | 10.65 ± 1.75 | 10.01 ± 2.15 | 3.20 ± 3.43 | 5.10 ± 0.77 | 1.15 ± 0.17 | 9.31 ± 0.01 | 12.02 ± 1.46 |
| Llama 3.1 8B | FedOPT + LoRA | 3.42 ± 1.43 | 11.79 ± 0.80 | 11.21 ± 1.05 | 3.80 ± 2.36 | 5.50 ± 0.06 | 1.08 ± 0.12 | 9.40 ± 0.53 | 12.33 ± 0.76 |
| | FedCDA + LoRA | **43.45 ± 1.03** | 13.89 ± 0.37 | 13.15 ± 1.00 | 15.74 ± 3.93 | 7.24 ± 0.84 | **1.37 ± 0.49** | 12.35 ± 0.85 | 14.13 ± 1.47 |
| | DRIFT SP | 17.91 ± 1.03 | 14.10 ± 0.21 | **13.90 ± 0.87** | 23.35 ± 3.64 | 7.52 ± 1.38 | 1.26 ± 0.17 | 12.41 ± 0.76 | **15.14 ± 0.99** |
| | DRIFT MST | 35.26 ± 1.40 | **14.37 ± 0.01** | 13.86 ± 0.55 | **32.23 ± 1.22** | **8.35 ± 0.58** | 1.25 ± 0.38 | **12.70 ± 2.26** | 15.04 ± 1.84 |
| | FedAvg + LoRA | 2.35 ± 0.01 | 13.07 ± 0.01 | 13.57 ± 0.01 | 0.97 ± 0.11 | 7.37 ± 0.01 | 0.20 ± 0.04 | 9.75 ± 0.04 | 10.97 ± 0.02 |
| | FedProx + LoRA | 2.32 ± 0.04 | 12.72 ± 0.05 | 13.26 ± 0.04 | 0.01 ± 0.06 | 7.27 ± 0.08 | 0.11 ± 0.01 | 9.45 ± 0.01 | 10.82 ± 0.20 |
| Qwen 2.5 7B | FedOPT + LoRA | 2.54 ± 0.01 | 13.19 ± 0.04 | 13.74 ± 0.02 | 1.07 ± 0.01 | 7.55 ± 0.01 | 0.13 ± 0.01 | 9.83 ± 0.25 | 11.07 ± 0.71 |
| | FedCDA + LoRA | 10.70 ± 1.63 | 14.14 ± 0.03 | 15.22 ± 0.23 | 12.74 ± 2.63 | 8.26 ± 0.06 | 0.48 ± 0.01 | 10.96 ± 0.25 | 13.10 ± 0.71 |
| | DRIFT SP | **39.13 ± 6.44** | 15.84 ± 0.03 | 17.59 ± 0.31 | 14.36 ± 0.02 | 10.58 ± 0.40 | 0.21 ± 0.01 | 12.52 ± 0.30 | **14.16 ± 0.55** |
| | DRIFT MST | 31.24± 3.61 | **16.53 ± 0.54** | **18.48 ± 0.49** | **17.76 ± 0.23** | **10.91 ± 0.82** | **2.24 ± 0.74** | **12.82 ± 0.34** | 13.91± 0.12 |

child nodes. As our node evaluator model, we used Deepseek R1 32B (DeepSeek-AI, 2025) and created preference datasets using CPO (Zhang et al., 2024). In addition to the two DRIFT variants, DRIFT MST and DRIFT SP, we used four well-established FL baselines FedCDA (Wang et al., 2024b), FedOPT (Reddi et al., 2021), FedProx (Li et al., 2020), and FedAvg (McMahan et al., 2017). However, we augmented each baseline strategy to be its LoRA (Hu et al., 2022) equivalent. We used Llama 3.1 8B (Grattafiori et al., 2024) and Qwen 2.5 7B (Yang et al., 2024) as our base models and conducted 50 FL rounds, in which each client was assigned its own dataset. To find the Shortest Path, we used Dijkstra's algorithm (Dijkstra, 1959), and for Minimum Spanning Tree we used Kruskal's algorithm (Kruskal, 1956). Note that these algorithms can be easily substituted for other graph-search algorithms. All experiments were conducted using three random seeds with 3 Nvidia-A100 GPUs. Details on hyperparameters, datasets, and prompts are provided in Appendix A, B, and D.1.

## 7.1 Evaluation Methods

We used three evaluation approaches to assess the quality of outputs generated by the models. Using the $N$ sample strategy, we sampled $N$ generations from trained models and measured the *Success Rate* with the detailed explanation of the solution provided in the test set. The objective behind using Success Rate is to demonstrate the effectiveness of each client's ability to generate high quality outputs, as each client is trained on high quality intermediate thoughts using CPO (Zhang et al., 2024). Second, we measured the *Accuracy* between the answer generated by the model and the final answer in the test set. Lastly, we evaluated the best *Reward* achieved by the trained model under each FL method. The choice of model reward is driven by our training objective, which is to optimize model performance on high-preference generations, $s_{i-1}^w = z_1^w, z_2^w, .., z_{i-1}^w$, without drastically deviating from the base models. Specifically, in DPO (Rafailov et al., 2023) the reward is defined as $r(x, z) = \beta \log \frac{\pi_x(z|x)}{\pi_{\text{ref}}(z|x)} + \beta \log Z(x)$. As CPO is an extension of DPO, the reward provides a signal on the model's ability to align its generation with the high preference thoughts. Therefore, an increasing reward reflects how well the model is able to align its generations with high preference generations compared to low preference generations.

## 7.2 Results and Discussion

Table 2, Figure 6, and Figure 7 summarize the best rewards achieved by DRIFT, on the evaluation datasets, compared to the baseline methods. On average, for Llama 3.1 8B, one of the two variants of DRIFT outperformed the baseline methods on 6 out of 8 datasets. On CSQA and PUBMEDQA, DRIFT SP on average generated 23% and 19% higher rewards. For COSE, MATHQA, MEDMCQA, and PIQA, DRIFT MST on average generated 21%, 3.2x, 36%, 6% higher rewards, respectively. On AQUA and MEDQA, FedCDA + LoRA produced the best performance. Similarly, for Qwen 2.5 7B, DRIFT outperforms other baselines across all datasets. Specifically, on AQUA and PUBMEDQA, DRIFT SP generated 6.8x and 22% higher rewards, whereas on COSE, CSQA, MATHQA, MEDMCQA, MEDQA, and PIQA, DRIFT MST on average produced 22%, 29%, 3.3x, 41%, 4.3x, 27% higher rewards, respectively.

Table 3 summarizes the success rate of each FL method. DRIFT consistently outperformed on all eight datasets. Specifically, compared to the baseline methods, on COSE, CSQA, MATHQA, MEDQA, and PIQA,

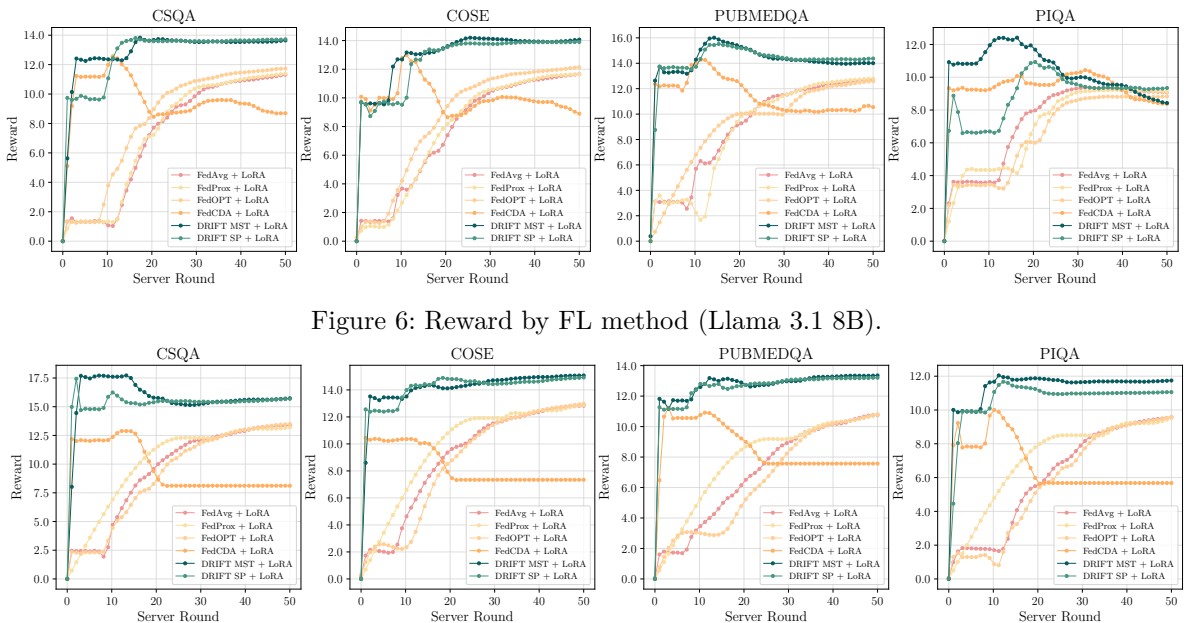

Figure 6: Reward by FL method (Llama 3.1 8B).

Figure 7: Rewards by FL Method (Qwen 2.5 7B).

DRIFT SP on average has a 3.1% higher success rate, whereas on AQUA, MEDMCQA, and PUBMEDQA, DRIFT MST has a 3.4% higher success rate. Similarly, we measured accuracy on the test sets of COSE, CSQA, MEDMCQA and PIQA and present the results in Table 4. DRIFT outperforms the baselines on all four datasets with DRIFT SP producing the best accuracy, closely followed by DRIFT MST. Relative to baselines, DRIFT SP, on average, produced 12.3%, 19.4%, 7.7%, and 11.2% higher accuracy on COSE, CSQA, MEDMCQA, and PIQA, respectively.

Both DRIFT variants enable clients to maintain their local characteristics while benefiting from custom aggregation with other clients. This allows each client to enhance its performance on the local data distribution by producing high-preference outputs resulting in improved success rate and accuracy. We illustrate this through a case study on the PUBMEDQA dataset in Figure 8. An additional case study on the CSQA dataset is provided in Appendix C, Figure 11. It is also evident from the results that higher rewards correspond to improved generation quality coinciding with improved success rate and accuracy.

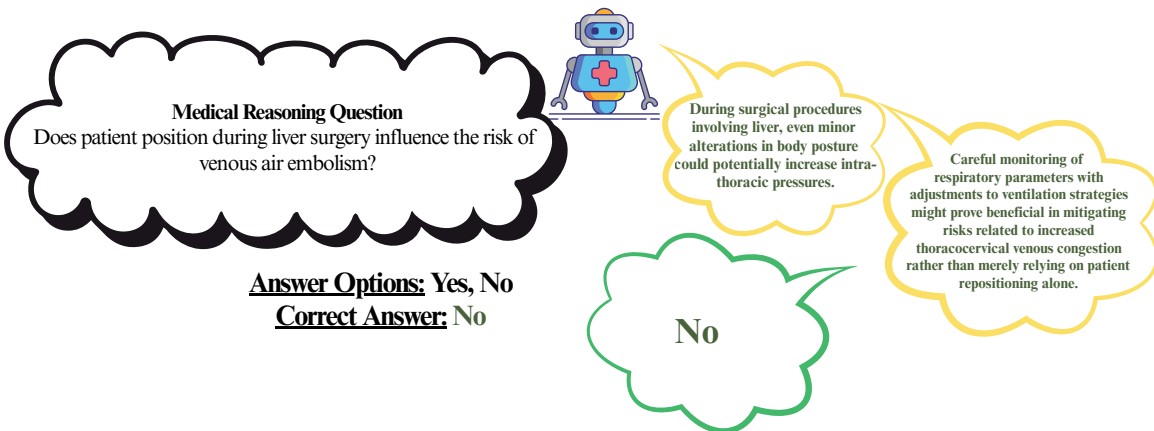

Figure 8: Case Study: green and yellow clouds show the output generated from a client model trained on PUBMEDQA.

Table 3: Success rate by FL method.

| Model | Method | Success Rate% | | | | | | | |
|---|---|---|---|---|---|---|---|---|---|
| | | AQUA | COSE | CSQA | MATHQA | MEDMCQA | MEDQA | PIQA | PUBMEDQA |
| | FedAvg + LoRA | 42.09 ± 0.16 | 90.35 ± 1.39 | 75.80 ± 3.04 | 42.26 ± 0.11 | 51.27 ± 0.18 | 78.76 ± 0.74 | 82.08 ± 0.20 | 46.74 ± 0.29 |
| | FedProx + LoRA | 42.10 ± 0.77 | 89.74 ± 2.27 | 76.45 ± 1.64 | 42.64 ± 0.80 | 50.87 ± 1.40 | 80.81 ± 1.10 | 84.41 ± 0.94 | 46.69 ± 0.51 |
| Llama 3.1 8B | FedOPT + LoRA | 40.58 ± 4.28 | 89.04 ± 0.22 | 75.34 ± 2.47 | 39.51 ± 6.59 | 51.05 ± 4.76 | 79.63 ± 0.95 | 84.12 ± 0.81 | 46.62 ± 0.47 |
| | FedCDA + LoRA | 40.40 ± 0.30 | 89.86 ± 0.58 | 74.97 ± 1.77 | 41.55 ± 0.46 | 51.81 ± 1.93 | 74.81 ± 1.67 | 79.35 ± 2.54 | 46.81 ± 0.21 |
| | DRIFT SP | 42.13 ± 0.12 | **92.34 ± 0.91** | **77.31 ± 0.32** | **43.29 ± 0.81** | 52.48 ± 0.23 | **81.07 ± 2.36** | **84.86 ± 1.56** | 47.21 ± 0.44 |
| | DRIFT MST | **42.55 ± 0.28** | 90.51 ± 3.05 | 76.66 ± 1.44 | 42.40 ± 0.72 | **54.23 ± 3.03** | 77.91 ± 1.18 | 81.77 ± 1.48 | **47.40 ± 0.28** |
| | FedAvg + LoRA | 40.30 ± 0.01 | 94.60 ± 0.03 | 81.43 ± 0.40 | 40.71 ± 0.31 | 52.22 ± 0.67 | 82.58 ± 0.15 | 86.71 ± 0.04 | 46.58 ± 0.04 |
| | FedProx + LoRA | 41.12 ± 0.66 | 94.87 ± 0.02 | 80.43 ± 0.43 | 40.91 ± 0.45 | 52.10 ± 0.62 | 82.01 ± 2.00 | 86.51 ± 0.44 | 46.74 ± 0.30 |
| Qwen 2.5 7B | FedOPT + LoRA | 40.47 ± 0.78 | 94.72 ± 0.17 | 80.77 ± 0.72 | 41.40 ± 0.04 | 51.80 ± 0.41 | 82.70 ± 1.14 | 87.80 ± 0.42 | 46.80 ± 0.41 |
| | FedCDA + LoRA | 41.10 ± 0.44 | 94.70 ± 0.02 | 80.30 ± 0.20 | 41.70 ± 0.80 | 51.20 ± 0.64 | 81.75 ± 1.70 | **92.24 ± 1.40** | 46.77 ± 0.29 |
| | DRIFT SP | **41.21 ± 0.56** | **95.54 ± 0.16** | 81.80 ± 1.13 | **41.71 ± 0.58** | 53.00 ± 0.28 | 82.47 ± 1.12 | 87.30 ± 1.46 | **47.41 ± 0.28** |
| | DRIFT MST | 40.91 ± 0.49 | 95.30 ± 0.33 | **82.21 ± 0.74** | 41.30 ± 0.31 | **53.50 ± 0.78** | **83.19 ± 0.19** | 88.50 ± 0.41 | 47.36 ± 0.05 |

Table 4: Accuracy by FL method.

| Method | Accuracy% | | | |
|---|---|---|---|---|
| | COSE | CSQA | MEDMCQA | PIQA |
| FedAvg + LoRA | 77.34 | 55.17 | 71.80 | 59.97 |
| FedProx + LoRA | 74.97 | 48.58 | 71.87 | 59.85 |
| FedOPT + LoRA | 76.81 | 51.79 | 74.55 | 66.67 |
| FedCDA + LoRA | 75.17 | 44.38 | 67.92 | 54.63 |
| DRIFT SP | **85.42** | **59.67** | **77.03** | **67.02** |
| DRIFT MST | _78.81_ | _57.44_ | 73.58 | 56.97 |

## 7.3 Parameter Study

To analyze the impact of divergence threshold ($\delta$) for DRIFT SP, we scaled the number of clients to 16, and for varying values of $\delta$ ($\delta \in \{0.0, 0.2, 0.4, 0.6, 0.8, 1.0\}$), conducted 25 FL rounds each. Table 5, Figure 9, and Figure 10 summarize the best rewards achieved for each $\delta$. In our analysis, we find that a lower $\delta$ generally leads to higher performance. Specifically, for Llama 3.1 8B, on COSE, MATHQA, PIQA, and PUBMEDQA, $\delta = 0.0$, on average, generated a 3.2% higher reward compared to other $\delta$ values. This is attributed to the fact that, at a lower $\delta$, a source client only merges with the least divergent clients, allowing it to maintain its parameter distribution. This experimental result verifies our analytical findings. For MEDQA and MEDMCQA, $\delta = 0.8$, achieves the best rewards, however, only marginally better than the rewards achieved using lower $\delta$ values. A similar pattern holds for Qwen 2.5 7B, where a lower $\delta$ ($\delta \in \{0.0, 0.2, 0.4\}$) outperforms a higher $\delta$ on 7 out of 8 datasets. On COSE, CSQA, MEDMCQA, MEDQA, PIQA, and PUBMEDQA, $\delta = 0.0$ and $\delta = 0.2$, on average generated 33.3% and 14.2% higher rewards from the lowest and second best performing $\delta$ values.

## 7.4 Computational Analysis

To aid the analysis of computational burden, we conducted experiments on a varied number of clients using both DRIFT SP and DRIFT MST. Table 6 presents the average wall clock time required for local training and server aggregation. The server aggregation time includes the time needed for graph creation, graph search, and parameter aggregation. Our findings show that the computational burden borne by the server is marginal compared to the clients' local training; however, the computational cost increases almost linearly as the number of clients increases.

## 8 Related Works

Traditional centralized FL aims to minimize the aggregate loss among all clients based on full model training; it updates all parameters of a neural network. To balance computational demands and privacy concerns, FL has been extended to training foundation models, particularly LLMs (Wang et al., 2025; Zhang et al., 2025a; Tran et al., 2025; Mahmoud et al., 2025; Rao et al., 2024; Hou et al., 2024; Panchal et al., 2024; Peng et al., 2024a; Pan et al., 2024; Sun et al., 2024). However, client heterogeneity causes Non-IID clients to degrade global model performance (Huang et al., 2025; Mishchenko et al., 2025; Yashwanth et al., 2024; Wang et al., 2024b; Makhija et al., 2024; Dai et al., 2024; Huang et al., 2024b; Fanì et al., 2024; Huang et al., 2024a).

Table 5: Parameter study for divergence threshold ($\delta$). Each row shows the best rewards, for each dataset, achieved by DRIFT SP for varying $\delta$. Legend: ▢ Best ▢ Medium ▢ Low.

| Model | Divergence Threshold ($\delta$) | 0.0 | 0.2 | 0.4 | 0.6 | 0.8 | 1.0 |
|---|---|---|---|---|---|---|---|
| | AQUA | $3.22 \pm 0.71$ | $2.46 \pm 0.63$ | $2.82 \pm 0.55$ | $2.39 \pm 0.67$ | $2.25 \pm 0.51$ | $2.85 \pm 0.66$ |
| | COSE | $15.87 \pm 2.83$ | $15.27 \pm 3.12$ | $15.04 \pm 2.72$ | $14.95 \pm 2.91$ | $14.83 \pm 3.22$ | $14.91 \pm 3.24$ |
| | CSQA | $15.30 \pm 2.85$ | $15.15 \pm 3.00$ | $15.19 \pm 2.89$ | $15.24 \pm 3.17$ | $14.56 \pm 3.34$ | $14.65 \pm 2.94$ |
| Llama 3.1 8B | MATHQA | $16.23 \pm 0.79$ | $15.15 \pm 0.72$ | $15.84 \pm 0.77$ | $14.88 \pm 0.63$ | $15.35 \pm 0.72$ | $15.61 \pm 0.75$ |
| | MEDMCQA | $5.44 \pm 1.35$ | $5.73 \pm 1.40$ | $5.68 \pm 1.42$ | $5.72 \pm 1.41$ | $5.74 \pm 1.41$ | $5.29 \pm 1.43$ |
| | MEDQA | $1.44 \pm 0.39$ | $1.41 \pm 0.37$ | $1.04 \pm 0.27$ | $0.86 \pm 0.23$ | $1.52 \pm 0.43$ | $1.49 \pm 0.39$ |
| | PIQA | $10.53 \pm 2.45$ | $10.60 \pm 2.31$ | $10.47 \pm 2.44$ | $10.23 \pm 2.61$ | $10.59 \pm 2.35$ | $10.51 \pm 2.41$ |
| | PUBMEDQA | $12.07 \pm 2.99$ | $12.35 \pm 3.02$ | $12.17 \pm 3.22$ | $12.11 \pm 3.01$ | $12.17 \pm 3.23$ | $12.02 \pm 3.06$ |
| | AQUA | $3.61 \pm 0.64$ | $3.47 \pm 0.64$ | $3.71 \pm 0.68$ | $2.12 \pm 0.42$ | $2.03 \pm 0.42$ | $2.12 \pm 0.40$ |
| | COSE | $14.43 \pm 3.23$ | $14.13 \pm 3.36$ | $11.41 \pm 2.89$ | $10.24 \pm 2.48$ | $11.12 \pm 2.75$ | $10.12 \pm 2.55$ |
| | CSQA | $12.91 \pm 3.27$ | $13.02 \pm 3.20$ | $11.85 \pm 3.17$ | $10.83 \pm 2.50$ | $10.16 \pm 2.68$ | $9.77 \pm 2.53$ |
| Qwen 2.5 7B | MATHQA | $1.49 \pm 0.50$ | $1.40 \pm 0.62$ | $1.50 \pm 0.56$ | $1.50 \pm 0.52$ | $1.58 \pm 0.29$ | $1.38 \pm 0.25$ |
| | MEDMCQA | $5.05 \pm 1.35$ | $6.36 \pm 1.55$ | $5.55 \pm 1.36$ | $4.48 \pm 1.10$ | $4.24 \pm 1.26$ | $4.81 \pm 1.36$ |
| | MEDQA | $0.24 \pm 0.08$ | $0.17 \pm 0.10$ | $0.20 \pm 0.08$ | $0.15 \pm 0.11$ | $0.19 \pm 0.11$ | $0.10 \pm 0.13$ |
| | PIQA | $9.93 \pm 2.23$ | $10.79 \pm 2.35$ | $8.38 \pm 1.91$ | $7.05 \pm 1.68$ | $7.11 \pm 1.71$ | $7.41 \pm 1.92$ |
| | PUBMEDQA | $9.63 \pm 2.22$ | $10.43 \pm 2.29$ | $9.21 \pm 2.17$ | $8.34 \pm 1.97$ | $7.81 \pm 1.84$ | $8.97 \pm 2.12$ |

Figure 9: Best rewards by divergence threshold ($\delta$) for Llama 3.1 8B.

Figure 10: Best rewards by divergence threshold ($\delta$) for Qwen 2.5 7B.

While many of the recent centralized FL methods address heterogeneous client and data settings, they still rely on the assumption that aggregating all clients can potentially improve performance. In contrast, our method is an extension of recent developments and is a server-side implementation of an FL algorithm that aggregates the LoRA parameters of clients in a heterogeneous setting, primarily driven by the goal of minimizing divergences between clients. To highlight the relationship of our method with related work, we provide a brief overview of relevant FL methods and their application on foundation models, particularly in the context of privacy-preservation and distributed learning among Non-IID clients.

## 8.1 Federated Foundation Model Training

Federated training of Foundation Models (FMs) is motivated by privacy-preservation, distributed data sources, or resource-constrained environments. Beitollahi et al. (2025) extract features from foundation models to train parametric models and share these models with the server in a one-shot FL setup, primarily to reduce communication cost in resource-constrained settings. JianHao et al. (2024) devise FedLPP, as a method to only quantize and integrate LoRA parameters for efficient fine-tuning. FlexLoRA Bai et al. (2024) synthesizes full LoRA weights using SVD and dynamically adjusts LoRA ranks. MPFT Zhang et al.

Table 6: Wall clock time (in seconds) for different numbers of clients.

| Method | # Clients | 4 | 8 | 16 |
|---|---|---|---|---|
| DRIFT SP | Aggregation | 4.4 s | 10.2 s | 22.2 s |
| | Local Training | 65.3 s | 72.5 s | 73.3 s |
| DRIFT MST | Aggregation | 4.4 s | 10.3 s | 22.1 s |
| | Local Training | 68.3 s | 70.8 s | 72.6 s |

(2025b) is an FL fine-tuning framework that enhances in-domain and out-of-domain performance by generating client-specific prototypes used to train a global adapter; the global adapter is further fine-tuned during a local adaptation phase. FedDAT Chen et al. (2024a) uses knowledge distillation to fine-tune foundation models without centralizing data. FedPFT Peng et al. (2024b) enhances the adaptation of foundation models by compressing and aligning sub-models for improved gradient accuracy. FedAPT (Su et al., 2024) achieves strong performance in diverse domains while using considerably less data through adaptive prompt tuning.

## 8.2 Data and Model Heterogeneity

Various works address data and model heterogeneity by defining custom training objectives (Jiang et al., 2024; Xiang et al., 2024; Xie et al., 2024; Liu et al., 2024b). A seminal work addressing client heterogeneity is FedProx (Li et al., 2020), which adds a proximal term which acts as a regularizer on the local objective. FedCDA (Wang et al., 2024b) addresses this issue in a cross-round setting by selecting and aggregating local models that minimize divergence from the global model, whereas FedSAK (Liao et al., 2024) addresses heterogeneity in a multitask setting. Gao et al. (2024) achieves improved computational efficiency with fewer communication rounds. InCo Aggregation (Chan et al., 2024) uses internal cross-layer gradients to improve similarity, while FedCompass (Li et al., 2024) reduces model staleness and straggler delays with data and device heterogeneity.

## 9 Conclusion

In this paper, we devised a novel server-side centralized FL aggregation algorithm, DRIFT, which measures divergence from a source client to other participating clients, building a graph-based structure. DRIFT utilizes graph search algorithms to find a set of least divergent clients and aggregates them with a source client. Using LLMs and PEFT, we applied DRIFT to the problem of preference optimization for language generation on a diverse set of domains. Our experimental results showed a significant performance improvement from other FL baselines. In addition, we conducted a parameter study supplemented by analytical findings to analyze how varying the divergence from a source client to other clients impacts performance. Experimental results on computation burden indicate that the computational cost of aggregation, primarily incurred by the server, is marginal compared to the local training of clients, but increases almost linearly as the number of clients increase. In our future work, we aim to further improve this method by incorporating graph partitioning to lower the computational cost of graph search. We hope that our work encourages further research in leveraging the graph properties of clients in FL.

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

## A  Model Parameters

We use 8-bit quantization for both Llama 3.1 8B and Qwen 2.5 7B with LoRA ($r = 16$). The training and evaluation batch sizes are both set to 4 per device, for improved memory-management. The model is trained using the AdamW 8-bit optimizer Dettmers et al. (2021), which is a memory-efficient variant of AdamW, and it utilizes float 16 precision for faster computation with lower memory usage. Gradient clipping is applied with a maximum gradient norm of 1 to stabilize training and prevent exploding gradients. The learning rate is set at $5e - 4$, however, we experiment with lower rates ($2e - 6$, $2e - 7$). A cosine learning rate scheduler is employed and the warm-up is configured for 4 steps, allowing the model to ease into full learning. We set $\beta = 0.2$, representing the strength of the KL-divergence regularization term that balances reward maximization with staying close to a reference policy. Finally, the input sequence is bounded with a maximum length of 512 tokens and a separate constraint on the prompt portion set to 256 tokens, optimizing memory and performance during both training and inference. A complete setup of model training and parameters is provided in our code base.

## B  Datasets

Our experiments are based on 8 different datasets, including CSQA Talmor et al. (2019), COSE Rajani et al. (2019), AQUA Ling et al. (2017), MATHQA Amini et al. (2019), PIQA Bisk et al. (2020), PUBMEDQA Jin et al. (2019), MEDQA Jin et al. (2021), and MEDMCQA Pal et al. (2022). These datasets cover commonsense reasoning, physical commonsense reasoning, medical reasoning, and mathematical reasoning. Table 7 provides details regarding each dataset used for model training and evaluation. From each dataset, we take approximately 300 questions. For each question, we further generated reasoning trees, using ToT Yao et al. (2023), with 2 child nodes and a depth of 3. Through pruning, preference datasets are created for model training as described in CPO Zhang et al. (2024). All of our training data is released within our code base.

## C  Model Output

Figure 11 shows another example output on the COSE dataset. The output is generated from a client model specific to the COSE dataset. The question along with its answer choices are shown at the top. Model response shows the output generated by the model followed by the final answer. The correct answer is the label associated with the question in the dataset.

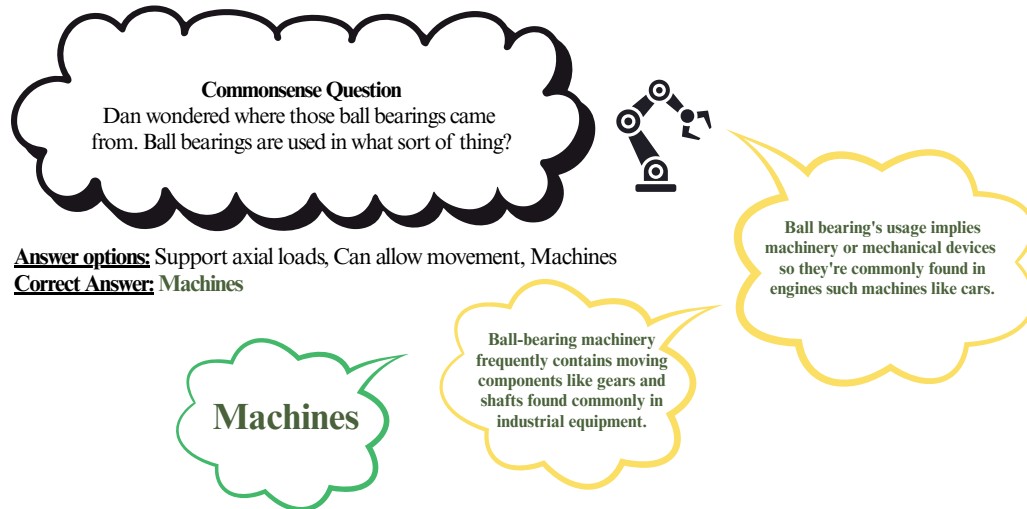

Figure 11: Case Study: output generated from a client model trained on COSE.

Table 7: Dataset details.

| Dataset | Reasoning Task | Description |
|---|---|---|
| AQUA | Mathematical Reasoning | Algebraic word problems, each with natural language rationales. Each problem includes a question, five answer options (A–E), a rationale, and the correct answer. |
| COSE | Commonsense Reasoning | CoS-E contains human-provided explanations for commonsense reasoning tasks. These explanations include both natural language descriptions and highlighted text annotations. |
| CSQA | Commonsense Reasoning | Commonsense Reasoning tasks with complex multi-hop inference. Each question has 5 potential answer choices. |
| MATHQA | Mathematical Reasoning | Diverse mathematical questions requiring symbolic and quantitative reasoning. The dataset is created by annotating the AQuA-RAT dataset using a newly introduced representation language. |
| MEDMCQA | Medical Reasoning | Multiple-choice questions (MCQs) covering medical knowledge. This dataset contains high-quality MCQs from AIIMS and NEET PG exams, spanning 2.4k healthcare topics across 21 medical subjects. |
| MEDQA | Medical Reasoning | This dataset comprises Multiple-choice questions (MCQs) sourced from the USMLE, reflecting professional medical board exam content. |
| PIQA | Physical Commonsense Reasoning | Tests physical interaction and intuitive knowledge of physics. This dataset was created to test the physical knowledge of models in Natural Language Processing. |
| PUBMEDQA | Medical Reasoning | Biomedical question answering based on PubMed abstracts. The goal of this dataset is to answer questions based on the three answer choices (yes, no, maybe) based on the given abstract. |

## D  Client Graphs

Figure 12 shows client graphs from different FL rounds. Each edge weight shows the divergence between two adjacent clients. Shortest path plots show the shortest paths from a source client to target clients. Minimum spanning tree plots show the minimum spanning tree of a client graph, as well as edges connecting a source client with its immediate neighbors.

### D.1  Sample Prompts

This section provides an example prompt, shown in Figure 13. The *Data Generation Prompt* provides a scenario in which the model must generate plausible and coherent responses to open-ended commonsense questions posed in CSQA. The goal is to simulate the reasoning process needed to answer these questions. Several example responses are provided for the initial question to illustrate the expected style and depth. Additionally, the *Value Prompt* is used for evaluation. It asks an evaluator model to score a generated thought from 1 to 10 based on how well it helps answer the question. This two-part structure helps both train and assess the model's ability to generate meaningful, contextually appropriate reasoning for common sense questions. Prompts with the same structure were used to generate data to create preference pairs for model training across all datasets used in our paper. For brevity, we provide one example. However, our code base includes prompts for each dataset.

## E  Environment and Libraries

We used Python as our main programming language along with NumPy Harris et al. (2020) and SciPy Virtanen et al. (2020) for array manipulation and scientific computing, Flower Beutel et al. (2020) for federated learning and for DRIFT implementation, Transformers Wolf et al. (2020) for working with Llama 3.1 8B and Qwen 2.5 7B, PEFT Mangrulkar et al. (2022) and TRL von Werra et al. (2020) for LoRA fine-tuning and preference optimization, PyTorch Paszke et al. (2019) for modeling, Matplotlib Hunter (2007) for generating figures, Pandas McKinney (2010) for data wrangling, and NetworkX Hagberg et al. (2008) for generating

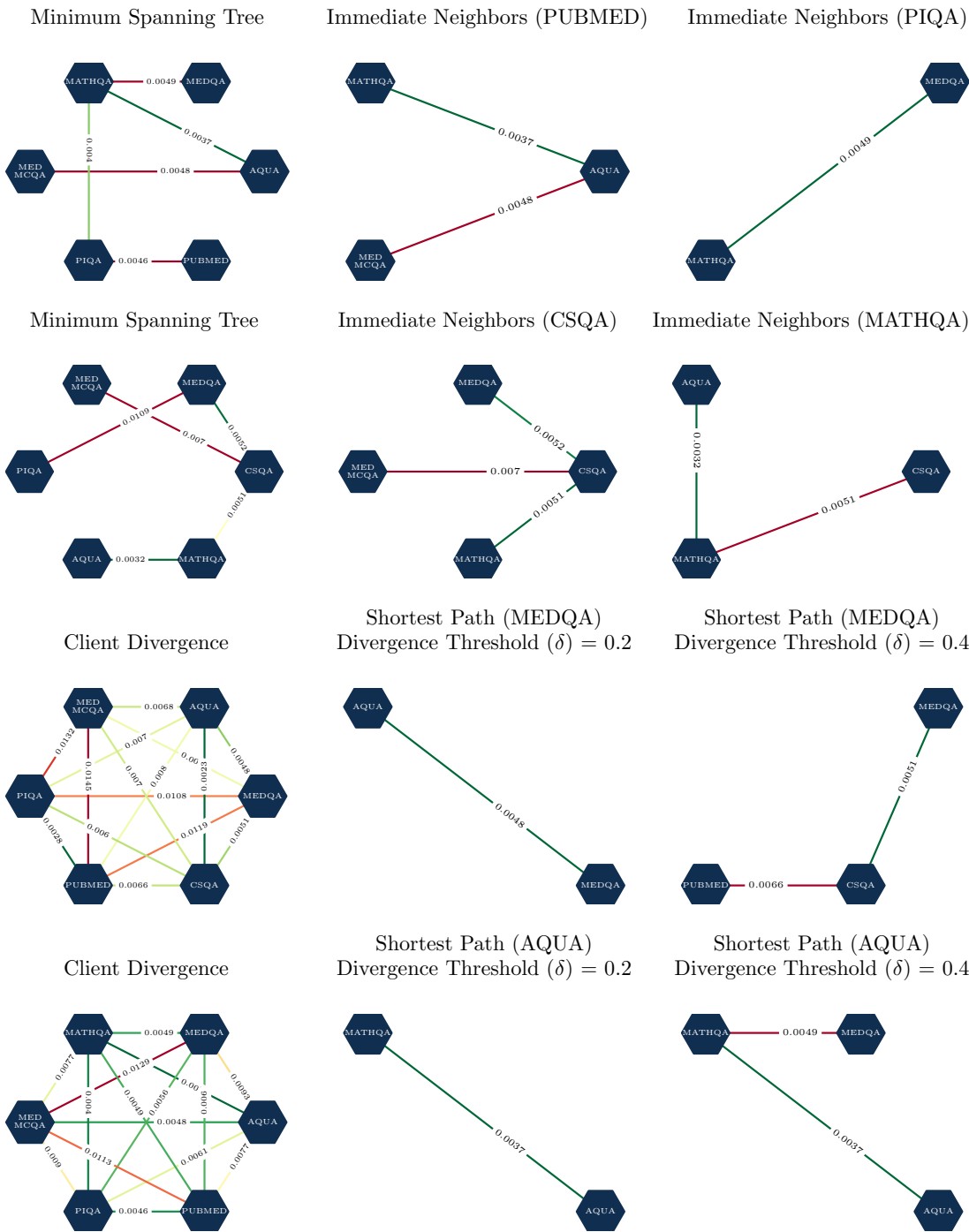

Figure 12: Client graphs from different FL rounds shows divergences between different clients.

graphs, saving graph data, and for conducting graph search. Specific versions for each library and framework along with additional requirements and their associated version numbers can be found in our code repository.

**Data Generation Prompt (CSQA)**:
Commonsense Question Answering focuses on developing systems capable of answering questions that require a deep understanding of everyday knowledge and human-like reasoning.
Generate a thought to answer to a Commonsense Question.
The current question (Human) and the thoughts (System) you have already generated are:
**Human:** Why does grooming require careful consideration?
**System:** Grooming requires careful consideration so personal appearance reflects positively upon oneself or others due to societal norms.
Grooming requires careful consideration so one doesn't present oneself unacceptably or poorly when interacting socially.
Grooming requires careful consideration since maintaining good hygiene promotes social acceptance and respect from peers and society through their perceptions about one's cleanliness.
Grooming requires careful consideration so one maintains personal hygiene and presents oneself appropriately in various social situations. `System`
Grooming requires careful consideration due to its impact on first impressions and social interactions.
Grooming requires careful consideration so one looks presentable before meeting others or attending special events like weddings and parties.
**Human:** The golfer was great at keeping a calm exterior as he finished up his final shots, but inside he was what because he knew he had won?
**System:** Winning can be thrilling and can result in exhilarating feeling.
**Human:** There are usually four strings on what popular instrument?
**System:** Instruments with strings can imply musical instruments. Some examples of musical instruments which contain strings include violins, banjos, guitars.
**Human:** You can read a magazine where while waiting for your transportation on rails to arrive?
**System:** If the mode of transportation involves rails then it could be a train.
**Human:** Why does grooming require careful consideration?
**System:**

**Value Prompt**:
Your task is to Score a Thought (between 1–10) which can help solve a Question. 1 being Worst and 10 being Best.
Question: {question}
Thought: {thought}
It is very important that your Score is a single integer value. Do not give me your reasoning. Only return an integer Score.
Score:

Figure 13: Example data generation prompt.

