# OpenReview forum: "United Yet Distinct: Domain Preservation via Divergence Reduction"
_TMLR — Rejected by TMLR_

### Review · Reviewer_Wjqe · 2025-12-10

**Summary Of Contributions:**

The paper is difficult to follow and lacks rigor in the presentation of its formulations and algorithm. The preliminary section is not ready for publication in its current form. In the part on federated learning, the authors should precisely describe the baseline methods. In the preference optimization section, the variables $\pi_{ref}$, $\beta$ are never defined, which prevents the reader from understanding the content.

I was also completely lost in Proposed Method section For example, what does the symbol written as \pau_i in the manuscript represent in the last line of Section 4.3? In addition, Equation (12) is unclear — is $\hat{p}$ meant to denote a set?

The analysis section is similarly difficult to understand. It is not clear what the authors aim to convey. Assumptions 1 and 4 appear without justification or explanation of their origin. The performance bound presented does not resemble a convergence bound; instead, it only concerns properties of the graph, which seems insufficient to support the claimed theoretical contributions.

Although the experimental section appears more complete, the overall writing quality falls short of the standard expected for a research paper. Significant revisions are necessary to improve clarity, precision, and coherence.

**Audience:**

Yes

**Audience Explanation:**

The topic on LLM may interest someone.

**Claims And Evidence:**

No

**Claims Explanation:**

The combination of missing definitions, unclear notation, weak structure, and imprecise explanations prevents the reader from understanding the technical contributions. Substantial rewriting is necessary before the paper can be properly evaluated.

**Requested Changes:**

Rewrite the paper in a clearer way

---

> ### Author Response · Authors · 2025-12-22
> **Response to Reviewer Wjqe**
>
> Thank you for taking the time to submit your review. To address the reviewer's critical concerns:
>
> 1. $\pi_{ref}$, $\beta$  are both well-defined in the Direct Preference Optimization (DPO) (1) and Chain of Preference Optimization (CPO) (2) objectives. Our paper builds on these concepts rather than provide a detailed survey. However, if the reviewer finds it necessary then we can incorporate more details regarding these methods in the appendix.
> 2. $\hat{p}$ is meant to denote a normalized set in Equation (12). In our revised version we will update the equation as follows:
>
> $\hat{p} = $\{$ \frac{|p_{i}|}{\sum_{i=1}^{t}|p_{i}|} : p_{i} \in p $\}$ $
>
>
> Please provide details regarding your concerns and where you have had difficulty. We will be happy to provide clarification.
>
>
> References:
> 1. Rafailov, R., Sharma, A., Mitchell, E., Manning, C. D., Ermon, S., & Finn, C. (2023). Direct preference optimization: Your language model is secretly a reward model. Advances in neural information processing systems, 36, 53728-53741.
>
> 2. Zhang, X., Du, C., Pang, T., Liu, Q., Gao, W., & Lin, M. (2024). Chain of preference optimization: Improving chain-of-thought reasoning in llms. Advances in Neural Information Processing Systems, 37, 333-356.

---

> > ### Comment · Reviewer_Wjqe · 2026-01-10
> >
> > >$\pi_{ref}$, $\beta$ are both well-defined in the Direct Preference Optimization (DPO) (1) and Chain of Preference Optimization (CPO) (2) objectives. Our paper builds on these concepts rather than provide a detailed survey. However, if the reviewer finds it necessary then we can incorporate more details regarding these methods in the appendix.
> >
> > The notation $\pi_{ref}$, $\beta$  did appear in the equations (2), but I still did not get the definitions. Can you point me to the sentences for the definitions, I might miss them somewhere?
> >
> > Some of my points that have not yet addressed:
> > >Assumptions 1 and 4 appear without justification or explanation of their origin. The performance bound presented does not resemble a convergence bound; instead, it only concerns properties of the graph, which seems insufficient to support the claimed theoretical contributions.
> >
> > This concern is similar to Reviewer fawS's point 4 and 5. I found the answer to assumption 1 in your response to Reveiwer fawS but not to assumption 4.
> >
> > >For example, what does the symbol written as \pau_i in the manuscript represent in the last line of Section 4.3?
> >
> > I recheck the revision and I am still confused on it. I think this question is also similar to one of Reviewer v4kb's questions. Let me be more precise: the equation $\hat \rho = \frac{\rho}{||\rho_i||_1}$ in the last line of Section 4.3 is unclear for me. What does this index $i$ mean? According to my understanding, $\hat \rho$ should be a vector, is it correct? If $\hat \rho$ is a normalized vector of $\rho$, the correct formule should be $\hat \rho = \frac{\rho}{||\rho||_1}$.
> >
> > Besides, in line 14 of Algorithm 1, we should probably point to this equation on $\hat \rho$ other than Equation 9. Speaking of Algorithm 1, lines 15 and 16 are unclear for me to understand how the aggregation happens. The last sentence of Section 4.1 mentions that "The client-specific aggregation under DRIFT MST is done by aggregating a given source client with the
> > target clients that are its immediate neighbor". But I did not find this interpretation in line 15. In addition, $k$ is the index of the client in the loop (for $k\in V$, line 14), what does it mean $\hat \Theta_k = \sum_{i=0}^k \hat \rho_k \Theta_k$ (line 15)? Shouldn't it be $\hat \Theta_k = \sum_{i=0}^{N(k)} \hat \rho_i \Theta_i$ where $N(k)$ is the neighbours of $k$ decided by the MST or SP? In line 16, I did not see the interest to update this $\hat \Theta$ as it will be rewrite immediately in the next round to an empty set. Where the algorithm utilize this $\hat \Theta$? Maybe I missed some parts but I still have the difficulties specially on understanding how the algorithm works.

---

> ### Author Response · Authors · 2026-01-12
> **Response to reviewer Wjqe**
>
> > The notation $\pi_{ref}$, $\beta$ did appear in the equations (2), but I still did not get the definitions. Can you point me to the sentences for the definitions, I might miss them somewhere?
>
> In the Direct Preference Optimization (DPO) paper please refer to section 3 Preliminaries. $\beta$ and $\pi_{ref}$ are defined on line 3 of the RL Fine-Tuning Phase. $\beta$ is a parameter which controls the deviation from the base reference policy, whereas $\pi_{ref}$ is the base reference policy.
>
> > This concern is similar to Reviewer fawS's point 4 and 5. I found the answer to assumption 1 in your response to Reveiwer fawS but not to assumption 4.
>
> This assumption also follows directly from the issue of Data Heterogeneity in Federated Learning. In a Non-IID setting it has been widely established that aggregating with more Non-IID clients leads to performance degradation. Assumption 4 is a restatement of this idea. Cong et. al (1) build on this idea. Similarly, Bhanbhro et.al (2) more explicitly state that they observe drop in performance as the number of clients increased in data heterogenous settings. Additional works which state and build on this idea include Wallach et. al (3) and Cao et. al (4).
>
> > I recheck the revision and I am still confused on it. I think this question is also similar to one of Reviewer v4kb's questions. Let me be more precise: the equation $\hat{\rho} = \frac{\rho}{||\rho_{i}||_{1}}$ in the last line of Section 4.3 is unclear for me. What does this index $i$ mean? According to my understanding, $\hat{\rho}$ should be a vector, is it correct? If $\hat{\rho}$ is a normalized vector of $\rho$, the correct formule should be $\hat{\rho} = \frac{\rho}{||\rho||1}$.
>
> $\hat{\rho}$ is indeed meant to be a normalized vector. We have posted a new revision to the paper and updated the formula in Equation (14) incorporating your feedback as: $\hat{\rho} = \frac{\rho}{||\rho||_{1}}$
>
> > Besides, in line 14 of Algorithm 1, we should probably point to this equation on other than Equation 9. Speaking of Algorithm 1, lines 15 and 16 are unclear for me to understand how the aggregation happens. The last sentence of Section 4.1 mentions that "The client-specific aggregation under DRIFT MST is done by aggregating a given source client with the target clients that are its immediate neighbor". But I did not find this interpretation in line 15. In addition, $k$ is the index of the client in the loop (for $k \in V$, line 14), what does it mean $\hat{\Theta_{k}} = \sum_{i=0}^{k} \hat{\rho_{k}}\Theta_{k}$ (line 15)? Shouldn't it  $\hat{\Theta_{k}} = \sum_{i=0}^{N(k)} \hat{\rho_{k}}\Theta_{k}$ be where $N(k)$ is the neighbours of $k$ decided by the MST or SP? In line 16, I did not see the interest to update this $\hat{\Theta}$ as it will be rewrite immediately in the next round to an empty set. Where the algorithm utilize this $\hat{\Theta}$? Maybe I missed some parts but I still have the difficulties specially on understanding how the algorithm works.
>
> - We have incorporated your feedback in our revision and line 14 of Algorithm 1 now points to Equation (14) which is the normalized weight vector.
> - Incorporating your feedback, we have updated Line 15 of Algorithm 1 as: $\Theta_{k_{t+1}} = \rho_{k}\Theta_{k} + \sum_{i \in s}\rho_{i}\Theta_{i}$. Here $S \subseteq K$ represent the neighbors of client k selected using MST or SP. This also remains consistent with Equation (5).
> - The idea behind $\hat{\Theta}$ was to explicitly state an updated set containing parameters for each client that participates in the current federated learning round. The loop is line 13 is over each client in $V$, which makes the set of vertices of the client graph, in the current federated learning round. Line 15 is the update for the given $k^{th}$ client in the loop. Line 16 is adding the updated weights to the set $\hat{\Theta}$. The set $\hat{\Theta}$ contains updated parameters for each client. The server broadcasts parameters to each client from this set.
>
> References:
> 1. Cong, Y., Zeng, Y., Qiu, J., Fang, Z., Zhang, L., Cheng, D., ... & Tian, Z. (2024). Fedga: A greedy approach to enhance federated learning with non-iid data. Knowledge-Based Systems, 301, 112201.
> 2. Bhanbhro, J., Nisticò, S., & Palopoli, L. (2024). Issues in federated learning: some experiments and preliminary results. Scientific Reports, 14(1), 1-15.
> 3. Wallach, E., Siler, S., & Deng, J. (2025). The More is not the Merrier: Investigating the Effect of Client Size on Federated Learning. arXiv preprint arXiv:2504.08198.
> 4. Cao, M., Zhang, Y., Ma, Z., & Zhao, M. (2022). C2S: Class-aware client selection for effective aggregation in federated learning. High-Confidence Computing, 2(3), 100068.

---

### Review · Reviewer_fawS · 2025-12-10

**Summary Of Contributions:**

This paper proposes DRIFT, a federated fine-tuning approach for LLMs under client heterogeneity (non-IID). The key contribution is a client-similarity–aware aggregation mechanism built on pairwise divergence between LoRA adapter updates (DRIFT for short).

DRIFT constructs a client graph using symmetric KL divergence over LoRA parameter distributions, selects personalized aggregation neighborhoods for each client via MST- or SP–based filtering with a divergence threshold, and performs inverse-distance weighted aggregation to obtain a customized global update for each client.

The submission provides theoretical analysis in Section 5 and experiment results in Section 7.

**Audience:**

Yes

**Audience Explanation:**

The considered personalization challenge in heterogeneous federated learning systems is an important area, e.g., how aggregation method can be developed, what criteria can be used to measure similarity between clients.

**Claims And Evidence:**

No

**Claims Explanation:**

1. It is not clear why symmetric KL divergence (in (7)) should be used: only a reference to Yao and Liu 2025 is not sufficient. The choice of Symmetric KL as the distance between LoRA parameters is only justified by non-negativity and symmetry. There is no empirical or theoretical evidence that SKL is a meaningful proxy of client similarity in this setting, nor comparison to alternative metrics (e.g., cosine, L2, Wasserstein)


2. Eq. (11) is not mathematically meaningful as written: although d_i are described as edge weights, the objective uses pairs $(d_{i−1}, d_i)$ without defining how such pairs contribute as scalars to the path cost. It is therefore unclear what “shortest path” is being optimized


3. Section 4.3 mechanically defines client weights as normalized inverses of the selected edge divergences, but provides no theoretical or empirical justification that this weighting scheme is appropriate or that it interacts meaningfully with the MST/SP objectives. The link between the graph-search formulation and the final aggregation rule is largely heuristic.


4. Assumption 1 ("In a Non-IID setting, client aggregation can degrade model performance") is merely a verbal statement that Non-IID aggregation can degrade performance, without any formal probabilistic or optimization statement. Combined with Theorem 3 (which only bounds path length, not performance such as accuracy, loss and reward), this is insufficient or misleading to support the claimed “performance bound”.

5. Assumption 4 and Theorem 5 yield the desired conclusion (shortest path aggregation improves performance), but neither side is justified: Assumption 4 is a vague verbal statement about performance, and Theorem 5 is a trivial restatement of the triangle inequality. As a result, the “analysis” is essentially circular and provides no nontrivial insight

6. The introduction mainly repeats generic motivations for federated LLM training (privacy, cost, Non-IID clients). It does not clearly articulate why graph-based aggregation via MST/SP over Symmetric KL of LoRA parameters is the right or necessary solution. As such, the motivation for the specific proposed method remains insufficiently articulated.

7. SOTA heterogeneity-aware FL baselines are missing. Only FedCDA is one of the heterogeneity-aware SOTAs. Other baselines include FedAvg, FedProx and Fed OPT, are not heterogeneity aware. Given the focus on heterogeneity-aware aggregation, it is important to compare against more recent personalized / cluster-based / heterogeneity-aware FL algorithms, especially those tailored to foundation models, to convincingly demonstrate the benefit of DRIFT

**Requested Changes:**

In addition to my concerns in "Are the claims made in the submission supported by accurate, convincing and clear evidence?", below please find my additional comments:

1. p is first defined in the first sentence of Section 4.2 as a set of weights, while it is redefined as a set of shortest paths before Eq (12).

2. $\omega, \Omega$ are defined in the proof of Theorem 3, but not defined in Theorem 3

3. Figure 4 and Figure 5 may need more clarifications and descriptions to cover the definitions of the graph, including nodes and edges.

4. In Eq (11), it is hard to understand the definition of $f(d)$, because $(d_{i-1},d_i)$ looks like an interval, rather than a scalar that joins the objective in the minimization.

---

> ### Author Response · Authors · 2025-12-22
> **Response to reviewer fawS**
>
> We sincerely thank the reviewer for their feedback. Please find our responses below. We will gladly incorporate your feedback in our revised version.
>
> 1: Symmetric KL Divergence as we describe on Page 4 in Equation (7), is additive KL Divergence from a source to a target distribution and vice versa. KL divergence has been extensively used to measure similarity between different probability distributions. Works which explicitly use Symmetric KL Divergence include Chen et. al (1), Andriamanalimanana et. al (2) where both works measure distance between distributions using Symmetric KL Divergence. Specifically in the context of model parameters, and because model parameters generally conform to Gaussian distributions, Huang et. al (3) propose using Symmetric KL Divergence to measure the distance between output layers of a DNN. Additional works which leverage Symmetric KL Divergence include Ruiz et. al (4) Pu et al. (5). If the reviewer desires, we can incorporate these works in our revised version.
>
> 2: Our objective is to find shortest paths between each pair of clients in the current federated learning round using the fully connected graph created by the server. The notation $\sum(d_{i-1}, d_{i})$ is meant to represent the sum of the weights between two constituent edges of a shortest path. However, we do recognize that without explicitly defining it as such, it can create confusion therefore in our revised version, we can provide this explanation in Section 4.2.
>
> 3, 6: Under a non-iid data assumption, naive averaging would directly aggregate each client; weighing each client equally. Using shortest path, in a fully connected graph, the edge costs/weights between the clients represent the locality of the clients. Clients would be aggregated as long as they are within the shortest path chosen for aggregation. Unlike naive averaging, inverse weights in this case would up-weight clients which are more similar to the source client.  Having more clients in the selected shortest path would be a case of exploration vs. having lesser clients in the selected shortest path would be a case of exploitation. Similarly, a MST of the client graph reveals the back-bond of the client graph in the most cost-effective manner. It is a subgraph which connects all clients using the minimal total cost in terms of the total edge weights. A minimum spanning tree therefore removes redundancy, revealing local linkage between clients (e.g., in the case of immediate neighbors). Inverse weights again up-weight similar client more compared to dissimilar clients.
>
> 4, 5: Assumption 1 is a classical Non-IID scenario in centralized federated learning. We stated it as such to avoid redundancy, however we provided references (Mishchenko et al., 2025; Hamidi & YANG, 2024; Vardhan et al., 2024; Li et al., 2020) which are works built to support this assumption. Similarly, many other works address the case of a Non-IID data setting, analogous to a heterogeneous setting. Theorem 3 is titled "Performance Bound" primarily because our work emphasizes that aggregation with more non-iid clients, in a heterogeneous setting, can lead to performance decline. This is not just expounded on in our analytical finding, in Theorem 3, but it is also supported with empirical evidence in Table 5.  Parameter study for divergence threshold ($\delta$).  Theorem 5, on the other hand, establishes a link between the graph search literature and our work. It specifically is designed to enable the reader bridge a gap between federated learning and the shortest path problem in graph search. The claim, which supports the proposed method, here is that the shortest path specifically in a federated learning set up enables aggregation with least divergent clients (client which are constituents of the shortest path), because any other path would incur a higher total cost in terms of aggregated edge weights.
>
> 6: To our knowledge, FedProx (10) and FedOpt (11), as devised, are both heterogeneity aware methods. FedProx (Section 3.2 - Proximal Term) is explicitly designed to help mitigate the effects of data and system heterogeneity. This is done by modifying the local training objective by adding a proximal term that keeps a client’s local model closer to the global model $\min_{w} \; F_k(w) + \frac{\mu}{2} \left\lVert w - w_t \right\rVert_2^2$. FedOpt accomplishes the task of minimizing heterogeneity by using an adaptive optimizer to aggregate the client objective (Section 3-Discussion).
>
> $\textbf{Response to Requested Changes}$
>
> 1: We use $p_i$ to define a path and $p$ to define a set of shortest paths. However, in our revision, we can denote all sets with capital letters.
>
> 2, 3: In our revision, we will ensure to incorporate definitions for both as well as detailed descriptions for Figure 4, 5.
>
> 4: $\sum(d_{i-1}, d_{i})$ is meant to denote the sum of the weights of the edges in the shortest path, but to improve readability, we can define it in Section 4.2.

---

> ### Author Response · Authors · 2025-12-22
> **References to the first comment.**
>
> References
> 1. Chen, Jiangning and Matzinger, Heinrich and Zhai, Haoyan and Zhou, Mi, Centroid estimation based on symmetric kl divergence for multinomial text classification problem, 2018 17th IEEE International Conference on Machine Learning and Applications (ICMLA) 1, 1174–1177 (2018).
>
> 2. Andriamanalimanana, Bruno and Tekeoglu, Ali and Bekiroglu, Korkut and Sengupta, Saumendra and Chiang, Chen-Fu and Reale, Michael and Novillo, Jorge, Symmetric kullback-leibler divergence of softmaxed distributions for anomaly scores, 2019 IEEE Conference on Communications and Network Security (CNS) 1, 1–6 (2019).
>
> 3. Huang, Z., Li, J., Siniscalchi, S. M., Chen, I. F., Wu, J., \& Lee, C. H. (2015, September). Rapid adaptation for deep neural networks through multi-task learning. In Interspeech (pp. 3625-3629).
>
> 4. Ruiz, F., \& Titsias, M. (2019, May). A contrastive divergence for combining variational inference and mcmc. In International conference on machine learning (pp. 5537-5545). PMLR.
>
> 5. Pu, Y., Wang, W., Henao, R., Chen, L., Gan, Z., Li, C., \& Carin, L. (2017). Adversarial symmetric variational autoencoder. Advances in neural information processing systems, 30.
>
> 6. Konstantin Mishchenko, Rustem Islamov, Eduard Gorbunov, and Samuel Horváth. Partially personalized
> federated learning: Breaking the curse of data heterogeneity. Transactions on Machine Learning Research,
> 2025. ISSN 2835-8856.
>
> 7. Shayan Mohajer Hamidi and EN-HUI YANG. Adafed: Fair federated learning via adaptive common descent
> direction. Transactions on Machine Learning Research, 2024. ISSN 2835-8856.
>
> 8. Harsh Vardhan, Avishek Ghosh, and Arya Mazumdar. An improved federated clustering algorithm with
> model-based clustering. Transactions on Machine Learning Research, 2024. ISSN 2835-8856.
>
> 9. Tian Li, Anit Kumar Sahu, Manzil Zaheer, Maziar Sanjabi, Ameet Talwalkar, and Virginia Smith. Federated
> optimization in heterogeneous networks. Proceedings of Machine learning and systems, 2:429–450, 2020.
>
> 10. Li, T., Sahu, A. K., Zaheer, M., Sanjabi, M., Talwalkar, A., \& Smith, V. (2020). Federated optimization in heterogeneous networks. Proceedings of Machine learning and systems, 2, 429-450.
>
> 11. Reddi, S., Charles, Z., Zaheer, M., Garrett, Z., Rush, K., Konečný, J., ... \& McMahan, H. B. (2020). Adaptive federated optimization. arXiv preprint arXiv:2003.00295.

---

### Review · Reviewer_v4kb · 2025-12-11

**Summary Of Contributions:**

In the manuscript, the federated learning (FL) problem is studied. The common issue with federated learning is data distribution shift. Each node in FL is supposed to have its own subset of data with a unique distribution. FL learning is motivated by privacy concerns.
Thus, learning becomes more complicated in the FL setting.
The non-IID nature of participating clients can degrade model performance.
Parameter Efficient Fine-Tuning (PEFT) enables adapting LLMs to downstream tasks with minimal parameter
additions and updates to their existing parameters. Preserving performance while learning from data in a distributed setting warrants the need for efficient training frameworks that can enable LLMs to learn from disparate data.
In this paper, authors design and propose a novel FL aggregation algorithm, Divergence Reduction in Federated Training (DRIFT), which accounts for the divergence between clients during model aggregation and disseminates
custom aggregated parameters back to each client. DRIFT measures the degree to which the PEFT parameters of the participating clients diverge and takes advantage of the graph-based structure implied by this divergence. DRIFT outperforms well-established baselines.
The code available in anon. repository.

**Audience:**

Yes

**Audience Explanation:**

The proposed algorithm concerns topics which are under active research now: federated learning, LLMs, parameter-efficient fine-tuning, data privacy.

**Claims And Evidence:**

Yes

**Claims Explanation:**

The main contributions as stated by authors are:

1) Design and implementation of a novel server-side centralized FL aggregation algorithm, for LLM
training.

The algorithm is introduced in Section 4, 6. Code is available in anon. repo.
Some points must be clarified, see Requested Changes.

2) Integrating our algorithm with cutting-edge training methods for LLMs and extending FL aggregation to graph-search algorithms.

Section 7.

3) Extensive experiments on a diverse set of 8 different natural language datasets with established FL
baselines using Llama 3.1 8B and Qwen 2.5 7B as base models.

Section 7.

**Requested Changes:**

1) In Equation (8) it is not clear which distribution is used. Thetas in LORA are added to weights. How to obtain probability distribution p then?
2) Some notation is non-trivial, for example \hat{p} in eq. (12) denotes a set. How a set can be divided by a scalar? A division must be element-wise.
3) Also, consider using a different notation style (capital letters, gothic, etc) for sets.
4) Is an index i is missing in \hat{\rho} in the of Section 4.3 ?
5) It is not clear, what happens if a dissimilarity graph is disconnected.
6) Pleas provide more details of purpose of "Success Rate" metric, why one should use it instead of Accuracy?

---

> ### Author Response · Authors · 2025-12-22
> **Response to reviewer v4kb**
>
> Dear reviewer, we are grateful for your well thought out review. Please see our response below. We will gladly incorporate your feedback in our revised version.
>
> 1. In Equation (8) it is not clear which distribution is used. Thetas in LORA are added to weights. How to obtain probability distribution p then?
>
> The probability distribution here denotes the probability distribution pertaining to each client's LoRA parameters. During the first Federated Learning round the LoRA parameters are initialized at the server and communicated to each client. Prior to the first local update, each client has identical LoRA parameters. The distribution of the each client's LoRA parameters becomes distinct as the clients start local training. The probability distributions in Equation(8) pertain to distinct clients between which KL divergence is measured. $\Theta$ here are the LoRA parameters, whereas $w$ represent each individual weight.
>
> 2. Some notation is non-trivial, for example $\hat{p}$ in eq. (12) denotes a set. How a set can be divided by a scalar? A division must be element-wise.
>
> Equation (12) is indeed meant to represent element-wise division of a set. We can make this more clear and update this equation in our revision as follows:
>
> $\hat{p} = $\{$ \frac{|p_{i}|}{\sum_{i=1}^{t}|p_{i}|} : p_{i} \in p $\}$ $
>
> 3. Also, consider using a different notation style (capital letters, gothic, etc) for sets.
>
> In our revision, we can update sets to be represented in capital letters.
>
> 4. Is an index i is missing in $\hat{\rho}$ in the of Section 4.3 ?
>
> Index i is not missing in Section 4.3. However, $\hat{\rho}$ defines a normalized vector of weights assigned to each client. We have also noted this in Table 1. Summary of main notational symbols.
>
> 5. It is not clear, what happens if a dissimilarity graph is disconnected.
>
> The server always creates a graph between all participating clients in the current federated learning round. The graph is created in such a way that each client connects with every other client as long as they are part of the participating clients. This forces the server to create a fully connected graph.
>
> 6. Please provide more details of purpose of "Success Rate" metric, why one should use it instead of Accuracy?
>
> To supplement Success Rate, we provided an accuracy comparison in Table 4. Accuracy by FL method. The main objective behind using Success Rate is to demonstrate the effectiveness of the task defined in our experimental setup. Each client is trained on intermediate thoughts $s_{i-1} = z_{1}, z_{2}, .., z_{i-1}$ which are deemed high quality thoughts using the Chain of Preference Optimization (CPO) (1) framework. Similar to Tree of Thoughts (ToT) (2) and CPO, Success Rate demonstrates the client's ability to generate a preferred intermediate thought (ideally the one which contains the correct answer). We can provide this explanation in our revised version as well.
>
> References:
> 1. Zhang, X., Du, C., Pang, T., Liu, Q., Gao, W., & Lin, M. (2024). Chain of preference optimization: Improving chain-of-thought reasoning in llms. Advances in Neural Information Processing Systems, 37, 333-356.
>
> 2. Yao, S., Yu, D., Zhao, J., Shafran, I., Griffiths, T., Cao, Y., & Narasimhan, K. (2023). Tree of thoughts: Deliberate problem solving with large language models. Advances in neural information processing systems, 36, 11809-11822.

---

### Decision · Action_Editor_Kev8 · 2026-02-01

**Recommendation:** Reject

**Audience:**

Yes

**Audience Explanation:**

The paper works on the relevant problem of designing a fine-tuning for federated learning of LLMs under client heterogeneity. Several members of the TMLR community (particularly those working on federated learning) would be interested in this paper.

**Claims And Evidence:**

No

**Claims Explanation:**

The paper works on the relevant problem of designing a fine-tuning strategy for federated learning of LLMs under client heterogeneity. It proposes a method that, at a high-level, is quite interesting, based on computing LoRA updates for each of the clients and then measuring divergences between the LoRA updates to identify client similarity.

However, the technical contribution of the paper feels very heuristic, with many design choices not fully justified. Moreover, and most importantly, the writing of the paper has significant room for improvement. Many claims, problem statements, and notation are mathematically imprecise and thus confusing. Even after reviewers pointed out the lack of clarity (e.g., in Section 4.2), the revised version is still below the required level of clarity for a technical paper. For example, Assumptions 1 and 4 are very strange and just verbal qualitative statements, without any formal technical assumptions.

Below, I also include final comments from reviewer fawS, regarding remaining issues after the revision. I believe these will be helpful for the authors in preparing a future major revision of this paper:

1. Compatibility of the weighting scheme with MST / shortest-path objectives (Section 4.3)

**Original concern:** why the client weighting scheme used in aggregation is theoretically or mechanistically compatible with the MST or shortest-path objectives, rather than being an ad hoc design choice.

**Authors’ response provides an intuitive explanation:** MST or shortest-path selection identifies similar clients, and inverse-divergence weights further emphasize those similar clients, with an exploration–exploitation interpretation.

**Remaining issue.** The response restates intuition but does not establish a formal or necessary connection between the graph objective (MST/SP) and the subsequent weighting rule. In particular, it remains unclear:

* why weighting is not already implicit in the path/tree selection,
* why inverse-divergence weighting is preferable to uniform or alternative schemes on the selected subgraph,
* or whether the proposed weighting follows from, or optimizes, the same objective as the graph construction.

Thus, the interaction between graph structure and aggregation weights remains heuristic rather than principled.

2. Assumptions used in the theoretical analysis (Assumption 1 / 4)

**Original concern:** a lack of well-defined, paper-internal assumptions that can legitimately serve as premises for the stated theorems, rather than informal references to “non-IID settings.”

**Authors’ response** characterizes these assumptions as classical non-IID scenarios and cites prior work showing that non-IID data can hurt performance.

**Remaining issue.** Citing prior observations about non-IID effects does not constitute a formal definition of the assumptions as used in this paper. The assumptions are still stated at a verbal level, without a precise probabilistic or optimization-level formulation, yet they are used to support subsequent theoretical claims. This lack of formalization limits the interpretability and rigor of the theoretical results.

3. Interpretation and naming of Theorems 3 and 5

**Original concern.** I questioned whether Theorem 3 legitimately constitutes a “performance bound,” and whether Theorem 5 provides nontrivial theoretical insight rather than restating properties of shortest paths.

**Authors’ response** explains that Theorem 3 is termed a performance bound because higher divergence empirically correlates with worse performance, and that Theorem 5 is intended to bridge graph search and federated learning.

**Remaining issue.** As stated, the theorems involve only graph quantities (e.g., path lengths, edge weights) and do not include any explicit performance metric or functional relationship to loss or accuracy. Consequently, the designation “performance bound” is not supported at the theorem level. Similarly, Theorem 5 follows directly from the definition of shortest paths and does not establish a nontrivial learning-theoretic result. The gap between theorem statements and the claimed performance interpretation remains unaddressed.

4. Motivation for the specific graph-based method

**Original concern** is on why graph-based aggregation via MST/shortest paths over symmetric KL is the right or necessary solution, as opposed to alternative heterogeneity-aware or personalization-based approaches.

**Authors’ response** discusses FedProx and FedOpt as heterogeneity-aware baselines.

**Remaining issue.** This response does not directly address the motivation for the proposed method itself. The question was not whether other methods handle heterogeneity, but why the particular graph-based formulation introduced in this paper is necessary or uniquely justified. This method-specific motivation remains insufficiently articulated.

5. Regarding the concern about baseline coverage, the rebuttal discusses FedProx and FedOpt as heterogeneity-aware methods, but does not address whether the experimental comparison sufficiently covers relevant personalized or clustering-based federated learning approaches, nor why the proposed graph-based aggregation is necessary relative to such methods.

**Resubmission Of Major Revision:**

The authors may consider submitting a major revision at a later time.